# Selective Role of TNFα and IL10 in Regulation of Barrier Properties of the Colon in DMH-Induced Tumor and Healthy Rats

**DOI:** 10.3390/ijms232415610

**Published:** 2022-12-09

**Authors:** Viktoria Bekusova, Tatiana Zudova, Ilyas Fatyykhov, Arina Fedorova, Salah Amasheh, Alexander G. Markov

**Affiliations:** 1Department of General Physiology, Faculty of Biology, Saint Petersburg State University, 199034 Saint Petersburg, Russia; 2Department of Veterinary Medicine, Institute of Veterinary Physiology, Freie Universität Berlin, 14163 Berlin, Germany

**Keywords:** barrier properties, colon, heterogeneity, cytokines, tumor necrosis factor alpha, interleukin 10, tight junction proteins, 1,2-dimethylhydrazine, colorectal cancer, rat

## Abstract

Recently it has been reported that the tumor adjacent colon tissues of 1,2-dymethylhydrazine induced (DMH)-rats revealed a high paracellular permeability. We hypothesized that the changes might be induced by cytokines. Colorectal cancer is accompanied by an increase in tumor necrosis factor alpha (TNFα) and interleukin 10 (IL10) that exert opposite regulatory effects on barrier properties of the colon, which is characterized by morphological and functional segmental heterogeneity. The aim of this study was to analyze the level of TNFα and IL10 in the colon segments of DMH-rats and to investigate their effects on barrier properties of the proximal and distal parts of the colon in healthy rats. Enzyme immunoassay analysis showed decreased TNFα in tumors in the distal part of the colon and increased IL10 in proximal tumors and in non-tumor tissues. Four-hour intraluminal exposure of the colon of healthy rats with cytokines showed reduced colon barrier function dependent on the cytokine: TNFα decreased it mainly in the distal part of the colon, whereas IL10 decreased it only in the proximal part. Western blot analysis revealed a more pronounced influence of IL10 on tight junction (TJ) proteins expression by down-regulation of the TJ proteins claudin-1, -2 and -4, and up-regulation of occludin only in the proximal part of the colon. These data may indicate a selective role of the cytokines in regulation of the barrier properties of the colon and a prominent role of IL10 in carcinogenesis in its proximal part.

## 1. Introduction

The large intestine is characterized by morphological and functional segmental heterogeneity [1,2,3]. The specificity of its two main different parts—the proximal and the distal colon, is manifested in many aspects, particularly, in development of right- or left-side colorectal carcinogenesis (CRC) [4,5,6,7,8,9,10,11,12,13]. Tumors more often appears in the proximal colon in adults [14], whereas in adolescents and young adults these are predominantly located in the distal colon [15]. A precancerous lesion of CRC as serrated adenoma was predominantly found in the distal part of the colon [16].

The most adequate to human CRC model of experimental carcinogenesis is 1,2-dymethylhydrazine (DMH)-induced CRC. DMH is a pro-carcinogenic agent that is activated in the liver and transported to the intestine by bile and blood. The mechanism of action of DMH is associated primarily with DNA methylation of the stem colonocytes which are located at the base of the intestinal crypts, with the subsequent development of colon adenocarcinomas [17,18,19]. The majority of colon tumors in the DMH-rats are located in the distal colon, while are less frequent in the proximal colon [17,18,20]. The cause of this preference remains unclear.

One of the reasons for the development of CRC is considered to be an imbalance between pro- and anti-inflammatory cytokines [21]. Pre-neoplastic lesions under DMH carcinogenesis may be connected with an increase in pro-inflammatory cytokines, especially tumor necrosis factor alpha (TNFα) [22]. It can be assumed that the different incidences of the colon may be associated with heterogeneity of pro- and anti-inflammatory cytokine production. The first group includes TNFα, while the second group includes interleukin 10 (IL10). There is no data available regarding the concentration of the cytokines in intestinal tissue in DMH-induced carcinogenesis.

CRC is accompanied by destroying the interactions between neighboring epithelial cells, cell dedifferentiation, loss of polarity, and metastasis. Impairment of the barrier properties of the colon and changes in the molecular composition of tight junctions (TJs) may lead to disturbance of the epithelial integrity [23].

TJs are composed of various transmembrane proteins, which belong to the MARVEL protein family, such as occludin and tricellulin, and the claudin family, which is divided in two groups according to their contribution to paracellular permeability [24]. Whereas some claudins as claudin-1, -3, -4 reduce epithelial permeability [25], others, particularly claudin-2, form pores and increase paracellular permeability for ions, water, and macromolecules [26]. One of the functions of TJs is to combine epithelial cells into unified sheets and maintain the integrity of the epithelial layer [27]. Under conditions of possible disintegration of TJs, the expression of claudins changes [28]. Therefore, one of the features of carcinogenesis, in particular in the colon, is a change in the expression of TJ proteins [29,30,31,32]. The physiological parameter for analysis of the destroying of the epithelial barrier function, consider the change in transepithelial resistance (TEER), short-circuit current (Isc) and paracellular permeability to some macromolecules such as sodium fluorescein [33].

Previously we have shown that DMH changed the intestinal permeability and induced alteration in expression of TJ proteins in the rat colon and IPEC-J2 cells [34,35]. We hypothesized that different incidence of the colon may be due to the different effects of the cytokines in carcinogenesis. TNFα decreased barrier properties of the colon ex vivo in Ussing chambers [36] and monolayers of epithelial cells in vitro [37,38,39,40]. IL10, on the contrary, had a protective effect, preventing disruption of the intestinal barrier function [41,42] and, perhaps, IL10 prevents the development of tumors in the proximal colon. Yet, the effects of cytokines on the barrier properties of the proximal and the distal parts of the colon have not been studied before. The aim of this study was to analyze the levels of TNFα and IL10 in the different segments of rat colon in DMH-induced carcinogenesis and to investigate the barrier properties of the main parts of the colon after intraluminal incubation with the cytokines in healthy rats.

## 2. Results

### 2.1. Study of the Level of TNFα and IL10 in Segments of Different Parts of the Colon during DMH-Induced Carcinogenesis

In the colon tissues, TNFα was different significantly only in the distal part. It was lower in the tumors compared to the control and tumor-adjacent tissues (Figure 1A). In comparison, IL10 was different only in the proximal part of the colon, where it was higher in tumors and non-tumor tissues compared to the control (Figure 1B).

### 2.2. Study of TEER, Isc and Paracellular Permeability to Sodium Fluorescein in the Proximal and Distal Parts of the Colon

The incubation with TNFα revealed an increase in paracellular permeability with a more pronounced effect in the distal part compared to controls (from 0.6 ± 0.1 to 1.4 ± 0.3·10^−4^ cm/s and 1.5 ± 0.3 to 4.5 ± 0.4·10^−4^ cm/s accordingly) (Figure 2A). TEER did not differ significantly compared to the control. Isc increased (from 15 ± 2 to 25 ± 3 µA/cm^2^) only in the distal part of the colon (Figure 2C).

In the proximal colon, incubation with IL10 revealed an increase in paracellular permeability (from 0.6 ± 0.1·to 1.2 ± 0.2·10^−4^ cm/s) (Figure 2A) and a decrease in TEER (from 170 ± 18 to 118 ± 12 Ω·cm^2^) (Figure 2B), whereas Isc did not show a significant change compared to the control.

### 2.3. Western Blotting in the Colon Tissues

The expression of TJ proteins, namely claudin-1, -2, -3, -4, occludin and tricellulin, after incubation with the cytokines was determined by immunoblotting, revealing that the level of TJ proteins significantly changed only in the proximal part of the colon (Figure 3). There were not significant differences in the distal part (Figure 4). TNFα reduced the level of claudin-1 (55 ± 22% vs. control, n = 4), while IL10 reduced the level of claudin-1 (32 ± 16% vs. control, n = 4), claudin-2 (36 ± 13% vs. control, n = 4) and claudin-4 (47 ± 17% vs. control, n = 4). At the same time IL10 increased the level of occludin (367 ± 209% vs. control, n = 4) (Figure 3). Other TJ proteins, namely, claudin-3 and tricellulin, did not show significant differences in all studied groups.

Thus, the cytokines induced a change in TJ proteins level only in the proximal part of the colon, and IL10 showed the most pronounced effect.

## 3. Discussion

Previously, it was found that the tumor-adjacent tissues in DMH-rats were characterized by a high paracellular permeability [34]. We hypothesized that these changes might be induced by cytokines. Therefore, we chose pro-inflammatory TNFα and anti-inflammatory IL10, which play opposite roles in the development of CRC, as well as regarding participation in the regulation of the barrier properties of the colon [36,37,38,39,40,41,42].

The role of cytokines in the development of CRC is ambiguous. An amount of TNFα in human blood serum was negatively correlated with proliferation of tumor cells [43]; however, increased circulating TNFα was associated with poor overall and cancer-specific patient survival [44]. In contrast, the level of IL10 has been shown to be positive correlated with the intensity of tumor proliferation and apoptosis [43]. However, it decreased in colon tissues with increased tumor invasion and lesion of lymph nodes at the late stages of CRC, corresponding to poor patient survival [45].

Most authors indicate that the concentration of TNFα in human blood serum in CRC is increased compared to the control [46,47,48,49,50]. The concentration of circulating TNFα was significantly higher compared to the control at all stages of CRC and showed highest values at the last, IV stage [46]. Expression of TNFα mRNA in tumor was significantly higher than in neighboring tissues [51]. Some authors reported that the level of IL10 was increased in human blood serum in CRC [49,50,52], while others showed that its concentration remained practically unchanged [53].

We investigated the cytokines level in the segments of the colon in DMH-rats in accordance with the mapping scheme we used earlier [34]. The study of the level of the cytokines into three types of segments—tumors, adjacent to tumors, and not adjacent to tumors, allowed us to analyze their concentration in the entire organ in detail, making a note of the possible influence in the tumor and tumor microenvironment on the colon tissues.

The level of TNFα was changed only in the distal part of the colon, it was lower in tumors compared to the control and tumor-adjacent tissues. We suppose that these results are explained by the features of our experimental model. Increased production of TNFα is usually associated with inflammation, and TNFα is a key cytokine that links inflammation and carcinogenesis [54]. DMH-induced carcinogenesis is not the direct result of an inflammatory process. It is believed that in the distal colon, histogenesis follows aberrant crypt foci-adenoma-carcinoma sequences, while in the proximal colon, carcinomas arise de novo without an intermediate stage of colon carcinogenesis [17]. Taking into account the inhibitory effect of TNFα on tumor cell proliferation [43] and that its pro-tumorigenic properties may rather be invoked by low chronic TNFα production than by an intensive outburst that activates reactive oxygen species and kills malignant cells [55], reduced concentration of TNFα in the tumors should have promoted tumor growth in the distal part of the colon. In fact, our hypothesis about the potential role of TNFα in the impairment of barrier properties of the colon and the possible negative effect of the tumor on neighboring tissues through the production of TNFα in DMH-induced carcinogenesis was not confirmed. That does not exclude its possible role in the disturbance of the barrier function with other types of CRC.

In our study IL10 was higher compared to the control in tumors, tumor-adjacent and not tumor-adjacent segments in the proximal part of the colon. It was most pronounced in the proximal not tumor-adjacent segments that significantly differed from the distal not tumor-adjacent segments. Thus, IL10 was significantly increased in all the segments of the proximal part of the colon in DMH-rats with right-side carcinogenesis. The studies revealed that IL10 can be a protective factor in animal CRC models [56,57]. Based on the fact that a decrease in IL10 in cancerous tissue is an independent risk factor for poor survival [45], elevated IL10 could have a protective effect in CRC. However, given that cytokines can play a dual role in tumor development—they can either participate in the suppression of carcinogenesis, or contribute to its progress [58], increased IL10 associated with tumors in the proximal part of the colon where tumors develop only in 20–25% of cases of CRC, could stimulate their development. Some studies suggest that IL10 serum levels are lower in the control group than in the CRC patients [59]. Patients in the fourth clinical stage of CRC have a higher level of serum IL10 when compared to lower stages, while a high serum concentration of IL10 correlates with poor survival of patients with CRC [60,61]. Additionally, IL10 overexpression was positively correlated with metastasis occurrence [62].

Previously, we conducted a segmental analysis of the barrier properties of the colon [63]. It was shown that the barrier properties were more pronounced in the proximal part of the colon compared to the distal part. We assume that the heterogeneity of the barrier properties of the colon and different effects of cytokines which are elevated in pathologies may determine how pathological processes develop and contribute to different incidences of CRC in two main parts of the colon.

Based on the literature, most studies of the barrier properties of the intestinal epithelium under cytokines action were carried out in vitro on cell cultures, with stimulation mainly from the basolateral side of the epithelium [37,64,65]. We stimulated the colon in situ from the apical side of the epithelium according to the approach we successfully used in our previous study [63]. We supposed that the addition of the cytokines to the mucosal side ensured their prolonged action and prevented their rapid enzymatic degradation by tissue cells.

We have shown that the four-hour prolonged action of cytokines on the colon of healthy rats led to the changes in the parameters of its barrier properties—a decrease in TEER, an increase in Isc, and an increase in paracellular permeability of the colon. Thus, under the action of TNFα and IL10 on the apical side of the epithelium, the barrier properties of the colon were reduced.

Regarding the main hypothesis that we checked, whether cytokines have the same regulatory effect on different parts of the colon, we analyzed barrier properties of the segments taken from its proximal and distal part. It was shown that TNFα changed the barrier properties mainly of the distal part of the colon, and IL10—only of the proximal one.

TJ proteins only showed changes in the proximal colon: TNFα reduced claudin-1, while IL10 decreased claudin-1, -2, -4, and increased occludin, demonstrating the most pronounced effect. The decrease of claudin-1 and -4, which increases the barrier properties of the colon, is consistent with the data obtained in Ussing chambers—an increase in paracellular permeability under the action of TNFα and IL10 and a decrease in TEER under the action of IL10 for the proximal colon segments. At the same time, increased occludin and decreased claudin-2 indicated an increase in the barrier properties of the proximal part of the colon and indicated a different effect of IL10 on the expression of TJ proteins in this region. Changes in TJ proteins expression only in the proximal part of the colon could indicate a greater sensitivity of this region to the action of cytokines and its ability to more pronounced molecular rearrangements.

Data on low TNFα in tumors in the distal part of the colon indicated that TNFα was probably not involved in the impairment of the barrier properties of the colon in DMH-induced carcinogenesis. Increased IL10 in the proximal segments of the colon of DMH-rats with proximal tumor location indicated that IL10 could play an important role in regulating the barrier properties in this colon segment. IL10 decreased the barrier properties and contributed to the restructuring of the TJ proteins only in the proximal part of the colon. Thus, IL10 altered intercellular interactions and could influence the development of CRC in the proximal part of the colon.

Our results have provided following novel findings: (1) cytokines have selective effects on the regulation of barrier properties of the colon: TNFα decreases barrier properties mainly in the distal part of the colon, while IL10 decreases the barrier properties only in the proximal one, (2) IL10 plays a prominent role in regulation of carcinogenesis and barrier function of the proximal part of the colon and may contribute to different incidences of CRC in the proximal and distal parts of the colon.

The heterogeneity of the cytokines production during cancer development and the selective effects of the cytokines on barrier properties of the proximal and the distal parts of the colon may contribute to different involvement of the colon in CRC.

## 4. Materials and Methods

### 4.1. Animals

Male Wistar rats for Experiment 1 were obtained from the Animal Laboratory of the I.P. Pavlov Institute of Physiology (Experiment 1, below: 5.1.) and from the vivarium of St. Petersburg University (Experiment 2, below: 5.2.). They were kept under a standard light/dark cycle (12 h light:12 h dark) at 22 ± 2 °C with ad libitum access to tap water and complete pelleted feed (Delta Feeds, BioPro, Novosibirsk, Russia). The studies were carried out in accordance with the guidelines of the FELASA [66] and approved by the Ethics Committee for Animal Research of St. Petersburg State University (Conclusion No. 131-03-1 dated 2 February 2021).

### 4.2. Chemicals

The cytokines were obtained from Sigma-Aldrich (Taufkirchen, Germany): TNFα from rat, recombinant, expressed in *E. coli*, IL10 human, recombinant, expressed in HEK 293 cells, HumanKine^®^.

## 5. Experimental Design

### 5.1. Experiment 1

Animals weighing 120–150 g (n = 20) were randomly subdivided into 2 groups—a control group (n = 6) and an experimental group (n = 14).

The rats in the control group were not exposed to the carcinogen, whereas the rats in the experimental group were administered 5 subcutaneous injections of DMH weekly at 21 mg/kg of body weight (each dose). DMH was obtained from Sigma-Aldrich (Tokyo, Japan). Six months after the first carcinogen injection, the rats were decapitated with a guillotine (Open Science, Moscow, Russia).

Mapping of the colon segments in the experimental group of DMH-induced rats depended on tumor location and was carried out according to the previously described method [34] (Figure 5). 

#### Study of TNFα and IL10 in Colon Tissues in DMH-Rats by Enzyme Immunoassay

The colon segments were immediately frozen at −80 °C and stored at this temperature until analysis. A total of 50–60 mg colon tissue samples were homogenized in 1 mL RIPA buffer with protease inhibitors (150 mM NaCl; 10 mM Tris-HCl, pH 7.4; 0.5% Triton X-100; 0.1% SDS), the suspension was sonicated and centrifuged (10,000× *g*, 5 min at 4 °C). In the supernatants, the concentration of total protein was determined. The cytokines in the samples were determined following the manufacturers’ instructions by sandwich ELISA and commercial reagent kits (High Sensitive ELISA Kit for IL10 and TNFα, Cloud-Clone Corp., Wuhan, China) using spectrophotometer SPECTROstar Nano (BMG LABTECH, Ortenberg, Germany). TNFα and IL10 concentration values were recalculated per mg of total protein.

### 5.2. Experiment 2

Animals weighing 280–380 g (n = 20) were randomly subdivided into four groups—one control group and three experimental groups with five animals in each group. The rats were anesthetized intraperitoneally with Zoletil 100 (Virbac, Carros, France, 100 mg/kg of the body weight). Narcosis was verified by the disappearance of the reaction to a painful stimulus (tail prick). An incision was made along the midline of the abdomen from the processus xiphoideus of the sternum in the distal direction. Two ligatures isolated the colon loop between the cecum and the anus, thus limiting the colon. Two tubes were inserted at the beginning and at the end of the loop, and the system was filled hermetically with the test solution. In the control group, the loop was luminally filled with Krebs-Ringer’s solution, while in the experimental groups it was filled with one of the cytokine solutions with 200 ng/mL TNFα and 100 ng/mL IL10. After 4 h of incubation the rats were decapitated, the large intestine was divided into proximal and distal segments, as described in our previous studies [34] and then investigated in Ussing chambers for 1 h (Figure 6).

#### 5.2.1. Electrophysiological Assay of the Colon Segments

Isc and TEER of the large intestine wall were studied according to the previously described protocol [67]. Briefly, the segments of the large intestine were mounted in Ussing chambers filled with Krebs-Ringer solution at 37 °C, which was maintained throughout the experiment, and permanently oxygenated with a mixture of 95% oxygen and 5% carbon dioxide. The Krebs-Ringer solution was composed as follows (mM): NaCl (119), KCl (5), MgCl_2_∙6H_2_O (1.2), NaHCO_3_ (25), NaH_2_PO_4_∙H_2_O (0.4), Na_2_HPO_4_∙7H_2_O (1.6), CaCl_2_ (1.2), and D-glucose (10). The short circuit current was recorded when the voltage was set at 0 mV. To evaluate TEER, we recorded voltage fluctuations when the current was 10 μA and calculated it using Ohm’s law: R = U/I (Ω). The size of the examined tissue was calculated by using the diameter of the slotted opening between the two chamber halves (4 mm) and was equal to 0.13 cm^2^. With regard to the size of the examined tissue, we adjusted the obtained transepithelial voltage values for 1 cm^2^ of tissue (Ω·cm^2^).

#### 5.2.2. Assessment of the Paracellular Permeability of the Colon Segments

To study the paracellular permeability of the intestine in the Ussing chamber, we added sodium fluorescein to the mucosal bathing solution at a final concentration of 100 μM. This concentration was determined from earlier published reports [68,69]. Thirty minutes after the experiment started, the serosal bathing solution was removed to analyze the concentration of the diffused sodium fluorescein. To assess the optical density of this solution, we used Cary Eclipse Fluorescence Spectrophotometer (Agilent Technologies, CA, USA). The excitation and emission wavelengths were 460 and 515 nm, respectively. The permeability coefficient (P_app_, cm/s) was calculated using the following equitation: P_app_ = (dQ/dt)/(A·C_0_), with: dQ/dt as the concentration of sodium fluorescein in the serosal bathing solution (mol/s), A as the size of the examined tissue (cm^2^), and C_0_ as the concentration of sodium fluorescein in the mucosal bathing solution.

#### 5.2.3. Western Blotting of Colon Tissues

TJ protein levels were analyzed in the colon segments, as described in detail earlier [70], and stain-free immunoblotting was performed, as described previously [71,72,73].

Briefly, tissues were homogenized in RIPA buffer (150 mM NaCl; 10 mM Tris-HCl, pH 7.4; 0.5% Triton X-100; 0.1% SDS) with protease inhibitor (Complete ULTRA Tablets, Mini; Roche, Mannheim, Germany), then centrifuged (15 min, 15,000× *g*, 4 °C), and a quantitative protein analysis using Thermo BCA assay kit (Thermo Fisher Scientific, Waltham, MA, USA) was performed by a spectrophotometer SPECTROstar Nano (BMG Labtech, Ortenberg, Germany). SDS buffer (Laemmli) was added to extracted proteins, and samples were loaded on 10% Stain-Free gels and electrophoresis was performed. Proteins from the gels were transferred to PVDF membranes with a 0.2 µm pore size (Bio-Rad, Hercules, CA, USA), which were first incubated with primary antibodies raised against claudin-1, -2, -3, -4, occludin or tricellulin and then visualized using secondary goat anti-rabbit and anti-mouse IgG antibodies and chemiluminescence reaction (Bio-Rad). The following antibodies were used: claudin-1 (#51-9000, Invitrogen, Carlsbad, CA, USA), claudin-2 (#32-5600, Invitrogen), claudin-3 (#34-1700, Invitrogen), claudin-4 (#36-4800, Invitrogen), occludin (#DF7504, Affinity Biosciences, Cincinnati, OH, USA), tricellulin (#48-8400, Invitrogen). The protein bands were detected and identified using Clarity Western ECL Substrate and the ChemiDoc XRS+ imager (Bio-Rad). Normalization of detected proteins was performed using the Image Lab 6.1 Software (Bio-Rad) to the total protein load measured in the membrane. The signal density in the control group was set as 100% (Appendix A).

### 5.3. Statistical Analysis

Statistical analysis was performed using the Anova group analysis in GraphPad Prism 8.4.3 (Graphpad Software Inc., San Diego, CA, USA). The data were analyzed using Mann–Whitney U-test. The results of the analyses are presented as mean ± standard error (M ± SEM). Statistically reliable differences were reported with a probability value of 95% (*p* < 0.05).

## Figures and Tables

**Figure 1 ijms-23-15610-f001:**
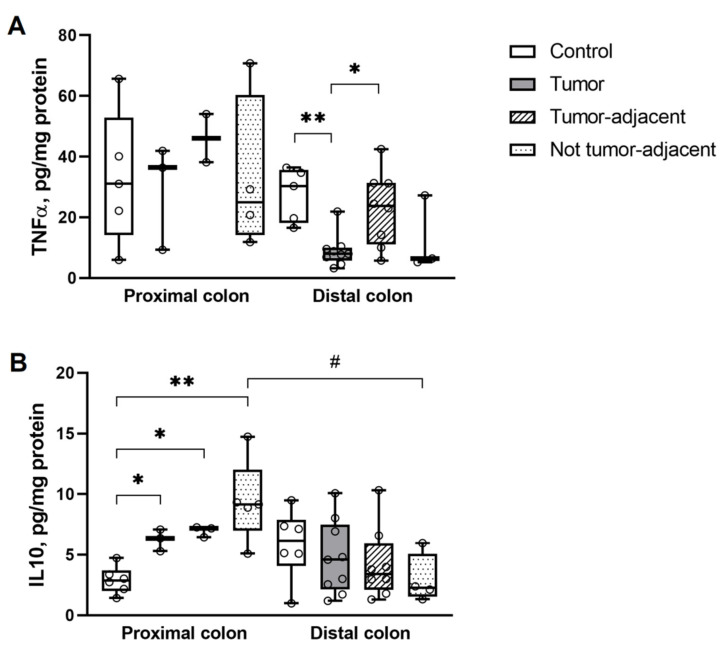
Elisa kit analysis of (**A**) TNFα and (**B**) IL10 in tumor, tumor-adjacent and non- tumor segments in the proximal and distal parts of the colon in DMH-rats. * *p* < 0.05, ** *p* < 0.005, # *p* < 0.05, Mann–Whitney U-test. The number of symbols corresponds to the number of samples.

**Figure 2 ijms-23-15610-f002:**
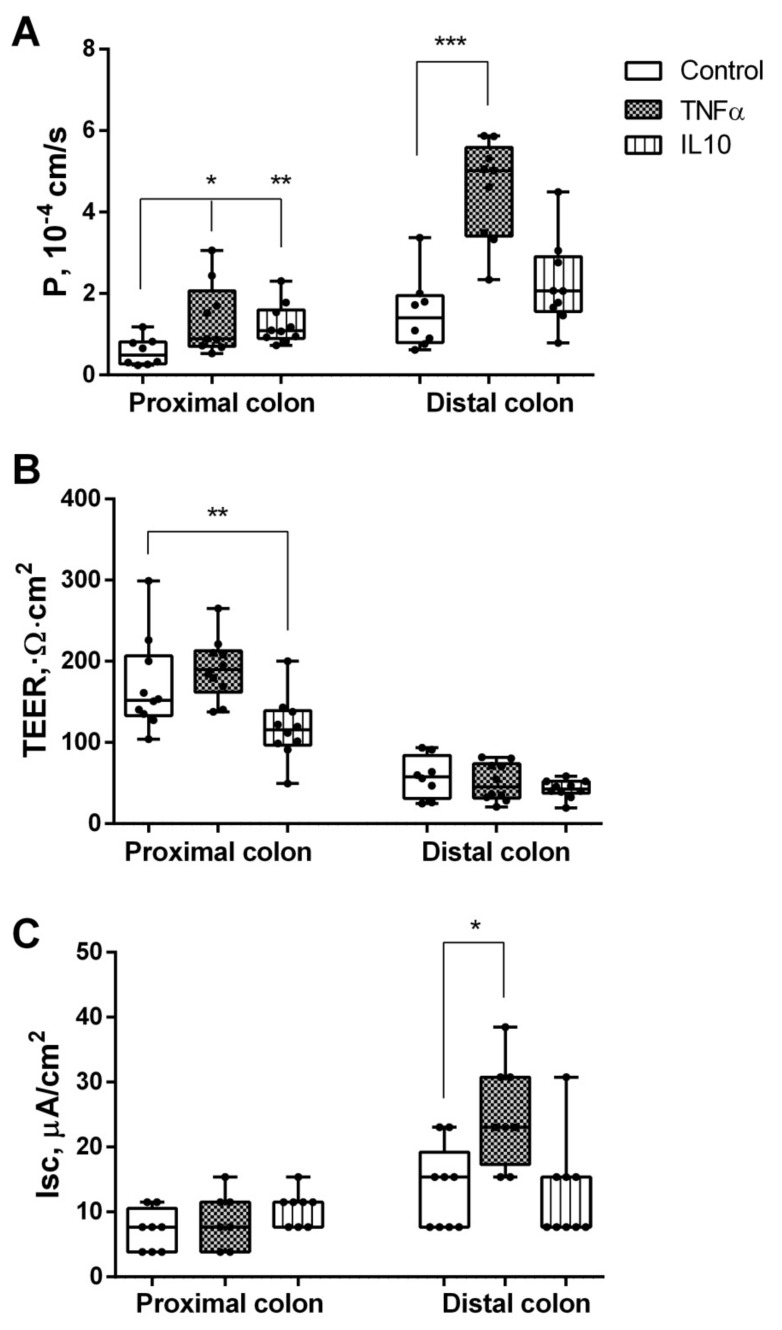
(**A**) Paracellular permeability for sodium fluorescein (P), (**B**) TEER, (**C**) Isc of the proximal and distal segments of the rat colon under the action of TNFα and IL10, * *p* < 0.05, ** *p* < 0.01, *** *p* < 0.001, Mann–Whitney U–test. The number of symbols corresponds to the number of samples.

**Figure 3 ijms-23-15610-f003:**
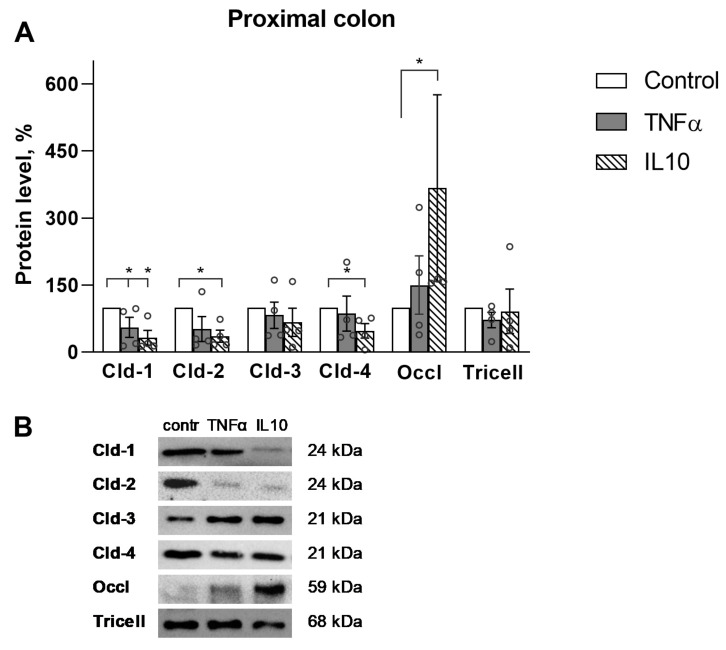
Western blot analysis of TJ proteins in the proximal colon. (**A**) Densitometric analysis revealed decreased Cld-1 under the action of TNFα, decreased claudin (Cld)-1, -2 and -4 and increased occludin (Occl) after IL10 action, Tricellulin (Tricell), * *p* < 0.05, Mann–Whitney U-test, n for each protein = 4. The values were normalized to total protein amount. The number of symbols corresponds to the number of samples. (**B**) Representative Western blot bands.

**Figure 4 ijms-23-15610-f004:**
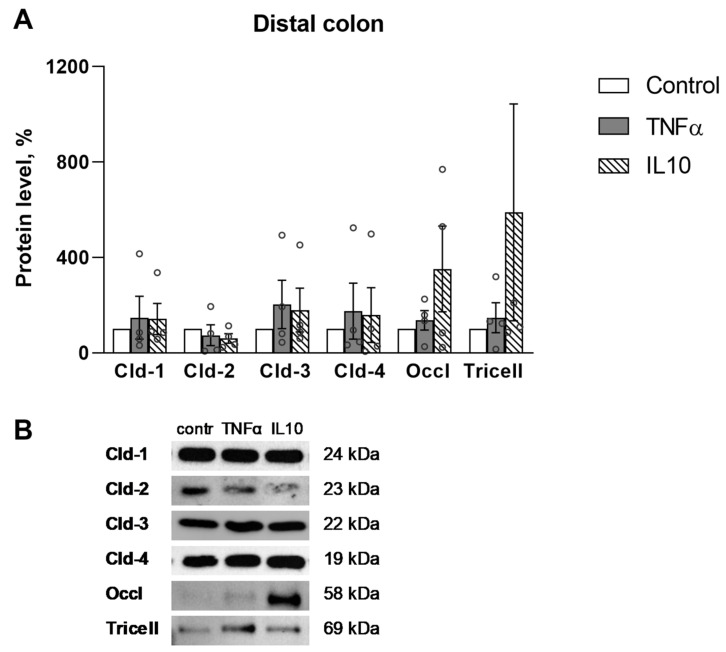
Western blot analysis of TJ proteins in the distal colon. (**A**) Densitometric analysis did not reveal any significant differences, *p* < 0.05, Mann–Whitney U-test, n for each protein = 4. The values were normalized to total protein amount. The number of symbols corresponds to the number of samples. (**B**) Representative Western blot bands.

**Figure 5 ijms-23-15610-f005:**
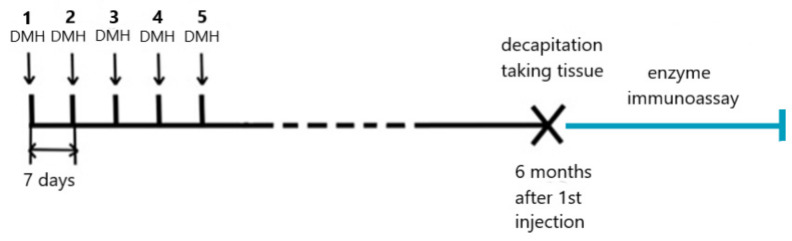
Design of experiment 1 (scheme).

**Figure 6 ijms-23-15610-f006:**
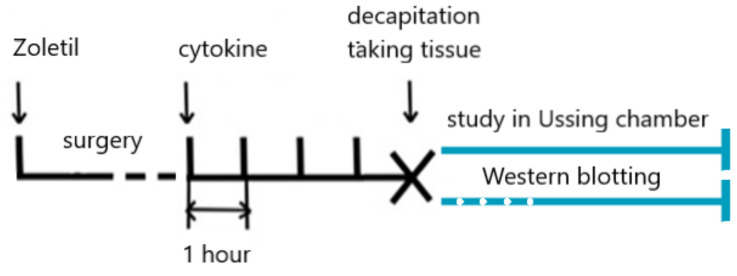
Design of experiment 2 (scheme).

## Data Availability

Date is contained in the article. The datasets analyzed in the study are available from the corresponding author upon reasonable request.

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
