# Peer review of "Selective Role of TNFα and IL10 in Regulation of Barrier Properties of the Colon in DMH-Induced Tumor and Healthy Rats"

_ijms, 2022, doi:10.3390/ijms232415610_

Round 1

Reviewer 1 Report

In this manuscript, the authors investigated TNFα and IL10 levels as well as the barrier properties of different colon segments in DMH-induced CRC rats through both in vivo and in vitro experiments. While the research is well-designed, there are still concerns that need to be addressed before the publication of this manuscript.

1.The anatomic distribution preference of CRC may vary depending on the ages or other factors of patient population. As reported by Siegel et al. (CA Cancer J Clin, 2020, PMID: 32133645), the overall incidence rates are highest for tumors in the proximal colon and lowest for those in the distal colon. In this manuscript, the authors claimed that tumors are more likely to develop in the distal part of colons (line49). Please provide references to support this statement. 

2. For the western blot results of colon tissues in figures 3&4, a loading control (e.g. GAPDH, Tubulin, β-actin) is required to ensure that the total protein amount loaded is equal across all samples.

3. For all abbreviations in the manuscript, their full names need to be provided when they first appear in the text. For example, TJ protein, TEER, Isc.

4. The title of the manuscript has a typo; it should be “Selective Role of TNFα and IL10 in……”  instead of " Selective Role of TNF and IL10 in……” (an “α” missing there).

5.The manuscript will greatly benefit from language editing by a native English speaker to eliminate any grammatic mistakes.

Author Response

Dear Sir/Madam,

We thank you for your insightful and helpful comments. We have addressed each of these with revisions to the text of the manuscript, accordingly.

Point 1: The anatomic distribution preference of CRC may vary depending on the ages or other factors of patient population. As reported by Siegel et al. (CA Cancer J Clin, 2020, PMID: 32133645), the overall incidence rates are highest for tumors in the proximal colon and lowest for those in the distal colon. In this manuscript, the authors claimed that tumors are more likely to develop in the distal part of colons (line49). Please provide references to support this statement.

Response 1: We have rewritten the introduction, added clarifying information to the text and presented the new references.

Point 2: For the western blot results of colon tissues in figures 3&4, a loading control (e.g. GAPDH, Tubulin, β-actin) is required to ensure that the total protein amount loaded is equal across all samples.

Response 2: Quantification of total protein using the Stain-Free technology that we used has advantages over household proteins for normalization, mainly due to the fact that this normalization does not depend on the expression of a single protein. Recent data show that the use of housekeeping proteins (β-actin or GAPDH) as a control may be even more inaccurate, since their levels may vary under certain conditions. When normalizing to a total protein, the intensity of all proteins in the membrane is used. Stain-Free gels contain trigalogen-containing compounds that modify tryptophans under the influence of UV radiation, creating a strong fluorescent signal proportional to the total amount of protein present. Quantification of total protein using the Stain-Free method serves as a reliable load control.

(https://www.ncbi.nlm.nih.gov/pmc/articles/PMC4791038/#R46)

We updated the file of supplementary materials (original images) with added original images of gels and total protein distribution and added some clarifying details in paragraph 5.2.3. Western blotting of colon tissues.

Point 3: For all abbreviations in the manuscript, their full names need to be provided when they first appear in the text. For example, TJ protein, TEER, Isc.

Response 3: We have corrected this, accordingly.

Point 4: The title of the manuscript has a typo; it should be “Selective Role of TNFα and IL10 in……” instead of " Selective Role of TNF and IL10 in……” (an “α” missing there).

Response 4: We have corrected this, accordingly.

Point 5: The manuscript will greatly benefit from language editing by a native English speaker to eliminate any grammatic mistakes.

Response 5: We have corrected it. If editing of English language and style would still be required, we will edit the manuscript using MDPI Author Services.

Reviewer 2 Report

ijms-2031339

Title: Selective Role of TNF and IL10 in Regulation of Barrier Properties of the Colon in DMH-induced tumor and healthy Rats

Corresponding author: Viktoria Bekusova

In general, the data in present studies are good and support the major conclusions of this manuscript. However, following issues need to be considered prior to considering the manuscript of publication.

1.    Because DMH plays a very important role in this experiment, sufficient description of DMH and previous research results using DMH should be discussed in the introduction.

2.    Abbreviations: The use of abbreviations when writing a paper has many advantages besides simplicity of expression. To use an abbreviation, first write the abbreviation in parentheses after the full name, and then use the abbreviation from Introduction to the final conclusion. In this paper, in most cases, the abbreviation is written first and then the full name is written in parentheses. In particular, because of the characteristics of IJMS, where Materials and Methods is arranged at the end of the paper, the original words and abbreviations are written in the order they are used from the introduction, and only when the abbreviation is used repeatedly, the abbreviation can be used until the conclusion. Since abbreviation for CRC is not additionally used in the abstract, so delete it.

3.    Line 17: Define TNF and IL here.

4.    Line 39: CRC in the abstract was abbreviated from colorectal cancer. Rewrite it here.

5.    Line 45: Re-write the sentence.

6.    Line 93: Define TJ.

7.    Figure 3A: Define Cld, Occl, and TRicell here it the figure legend.

8.    Materials and Methods section - When naming a particular chemical company, you must provide location information such as company name, city and/or state (abbreviation in the USA and Canada) and country. Once you have named a company with the information, you should only mention a company’s name thereafter.

9.    Line 238: Sigma Aldrich should be Sigma-Aldrich.

10.    Line 239: E. coli should be written in italic.

11.    Line 276: Delete a space between ml and TNF.

12.    Lines 283 and 310: Two forms of temperature units are used. Please stick on one of them. 4°C and 37 °C. The space between number and unit is the problem.

13.    Lines 258 and 301: When expressing the force of a centrifuge, it is good to distinguish 'g' from the unit of weight by italics.

14.    Reference section: Author should consult and peruse carefully recent issues of the journal, International Journal of Molecular Sciences, for format and style. The first letter of the title must be in upper case, and the rest must be in lower case. Examples: 2, 4, 7, 11, 16, 19, 21, 23, 33, 41, etc.

Overall, the manuscript can be considered to publication after major revision as indicated above.

Author Response

Dear Sir/Madam,

We thank you for your insightful and helpful comments. We have addressed each of these with revisions to the text of the manuscript, accordingly. If editing of English language and style would still be required, we will edit the manuscript using MDPI Author Services.

Point 1: Because DMH plays a very important role in this experiment, sufficient description of DMH and previous research results using DMH should be discussed in the introduction.

Response 1: We have rewritten the introduction, added clarified information to the text and support it with appropriate references. The links point to the articles that describe this model in detail.

Point 2: Abbreviations: The use of abbreviations when writing a paper has many advantages besides simplicity of expression. To use an abbreviation, first write the abbreviation in parentheses after the full name, and then use the abbreviation from Introduction to the final conclusion. In this paper, in most cases, the abbreviation is written first and then the full name is written in parentheses. In particular, because of the characteristics of IJMS, where Materials and Methods is arranged at the end of the paper, the original words and abbreviations are written in the order they are used from the introduction, and only when the abbreviation is used repeatedly, the abbreviation can be used until the conclusion. Since abbreviation for CRC is not additionally used in the abstract, so delete it.

  1. Line 17: Define TNF and IL here.
  2. Line 39: CRC in the abstract was abbreviated from colorectal cancer. Rewrite it here.
  3. Line 45: Re-write the sentence.
  4. Line 93: Define TJ.
  5. Figure 3A: Define Cld, Occl, and TRicell here it the figure legend.

Response 2-7: We have addressed and corrected all points, accordingly.

Point 8: Materials and Methods section - When naming a particular chemical company, you must provide location information such as company name, city and/or state (abbreviation in the USA and Canada) and country. Once you have named a company with the information, you should only mention a company’s name thereafter.

  1. Line 238: Sigma Aldrich should be Sigma-Aldrich.
  2. Line 239: E. coli should be written in italic.
  3. Line 276: Delete a space between ml and TNF.
  4. Lines 283 and 310: Two forms of temperature units are used. Please stick on one of them. 4°C and 37 °C. The space between number and unit is the problem.
  5. Lines 258 and 310: When expressing the force of a centrifuge, it is good to distinguish 'g' from the unit of weight by italics.
  6. Reference section: Author should consult and peruse carefully recent issues of the journal, International Journal of Molecular Sciences, for format and style. The first letter of the title must be in upper case, and the rest must be in lower case. Examples: 2, 4, 7, 11, 16, 19, 21, 23, 33, 41, etc.

Response 8-14: We have corrected all these single points, accordingly.

Reviewer 3 Report

The manuscript presents the results modulation of cytokines, but in my opinion, it is not a new approach.

Comments

The Authors should give a better justification for research in the introductory part

The Authors should add whole membranes in supplementary materials

The Authors should conduct the expression of genes respectively to study proteins

Please add a scheme/ graph of experiments with animals

The Authors should explain the doses of DMH used in this study

Author Response

Dear Sir/Madam,

We thank you for your insightful and helpful comments. We have addressed each of these with revisions to the text of the manuscript, accordingly.

Point 1: The manuscript presents the results modulation of cytokines, but in my opinion, it is not a new approach.

Response 1: We agree with you that investigation of effects of modulation of barrier properties by cytokines is not a new approach. The novelty of our work is the comparative study of: 1. effects of the cytokines in different (proximal and distal) parts of the colon, 2. concentrations of the cytokines in different segments of the colon in DMH-rats (please see also response 2).

Point 2: The Authors should give a better justification for research in the introductory part.

Response 2: We have rewritten the introduction to better represent the rationale of the study.

Point 3: The Authors should add whole membranes in supplementary materials.

Response 3: We have added images of the gels to each membrane in supplementary materials (Original images).

Point 4: The Authors should conduct the expression of genes respectively to study proteins.

Response 4: In planning this work, we focused on functional proteins that are well defined by the Western blot method. We agree that the combination of two approaches – Western blot and gene expression, gives the most reliable results. We thank you for your remark and will take it into account when planning further studies.

Point 5: Please add a scheme/ graph of experiments with animals.

Response 5: We added Figures 5 and 6 with animal experiment design disclosure to the Materials and Methods section.

Point 6: The Authors should explain the doses of DMH used in this study.

Response 6: We used 5 subcutaneous injections of DMH weekly at a single dose of 21 mg/kg of body weight. This dose was determined to be sufficient for the induction of colorectal cancer in Wistar rats (Pozharissky K.M., 1975). In this case, tumors occur mainly in the large intestine in 100% of Wistar rats 5-6 months after the first injection. The same dose, along with other possible doses, is mentioned in recent reviews of this model (references 17, 18 in the manuscript).

Reviewer 4 Report

Comments to the author:

The article “Selective Role of TNF and IL10 in Regulation of Barrier Properties of 2 the Colon in DMH-induced tumor and healthy Rats” (ijms-2031339) is a
preliminary research which aims to clarify the link between the levels of some cytokines and their effects on colon epithelial barrier in induced colorectal cancer tissue and in tumor adjacent colon tissues in mice. However, despite the interest of the topic, there are some major worrying concerns about the project and its conclusion. There are some serious problem in how the topic is presented and in the western blot experiments.
We definitely invite you to review again the assumptions and the results.
Here reported are
some suggestions to improve your manuscript.

1)    We strongly recommend you introduce a paragraph where you could explain what 1.2-dymethilgidrazine is and how it works, in order to ease people who do not know the mechanism of this substance to better understand your project.

2)    Why did you choose only TNFα and IL10 for your research? Please explain this in your introduction and support your information with literature data.

3)    We don’t think that normalize your western blots on total protein amount is scientifically correct. You must always normalize your protein with a housekeeping protein, like actin and hybridize it on the same membrane of your target. Then, all images must be reported in your raw data. Please, repeat all your western blots experiments following the indications given.

4)    Your conclusions turn out to be weak and confusing, not confirming your initial hypothesis. You must overhaul this paragraph and better draw the conclusion of this project an look for more full-bodied results.

Author Response

Dear Sir/Madam,

We thank you for your insightful and helpful comments. We have addressed each of these with revisions to the text of the manuscript, accordingly.

Point 1: We strongly recommend you introduce a paragraph where you could explain what 1.2-dymethilgidrazine is and how it works, in order to ease people who do not know the mechanism of this substance to better understand your project.

Response 1: We added clarified information to the introduction and support it with appropriate references. The links point to the articles that descript this model in detail.

Point 2: Why did you choose only TNFα and IL10 for your research? Please explain this in your introduction and support your information with literature data.

Response 2: We have rewritten the introduction and tried to better justify the project on the basis of literature data.

Point 3: We don’t think that normalize your western blots on total protein amount is scientifically correct. You must always normalize your protein with a housekeeping protein, like actin and hybridize it on the same membrane of your target. Then, all images must be reported in your raw data. Please, repeat all your western blots experiments following the indications given.

Response 3: Recent data show that the use of housekeeping proteins (β-actin or GAPDH) as a control may be inaccurate, since their levels may vary under certain conditions. When normalizing to a total protein, the intensity of all proteins in the membrane is used. Quantification of total protein using the Stain-Free technology that we used has advantages over household proteins for normalization, mainly due to the fact that this normalization does not depend on the expression of a single protein. Stain-Free gels contain trigalogen-containing compounds that modify tryptophans under the influence of UV radiation, creating a strong fluorescent signal proportional to the total amount of protein present. Quantification of total protein using the Stain-Free method serves as a reliable load control.

(https://www.ncbi.nlm.nih.gov/pmc/articles/PMC4791038/#R46)

We updated the file of supplementary materials (Original images) with added original images of gels and total protein distribution and added some clarifying details in paragraph 5.2.3. Western blotting of colon tissues.

Point 4: Your conclusions turn out to be weak and confusing, not confirming your initial hypothesis. You must overhaul this paragraph and better draw the conclusion of this project an look for more full-bodied results.

Response 4: We have rewritten introduction to clarify our initial hypothesis, added additional information to discussion and excluded one of the conclusions (number 2) that was only our guess in order to reconcile our results and conclusions.

Round 2

Reviewer 2 Report

ijms-2031339-v2

Title: Selective Role of TNF and IL10 in Regulation of Barrier Properties of the Colon in DMH-induced tumor and healthy Rats

Corresponding author: Viktoria Bekusova 

The revised manuscript has been greatly improved and has been very helpful for reading and understanding. However, some additional revisions are required as pointed out below.

1.   Line 3 (Title of the Article): Two words such as 'tumor' and 'healthy' should be written as 'Tumor' and 'Healthy'. 

2.   Abbreviations: The use of abbreviations when writing a paper has many advantages besides simplicity of expression. To use an abbreviation, first write the abbreviation in parentheses after the full name, and then use the abbreviation from Introduction to the final conclusion. In this paper, in most cases, the abbreviation is written first and then the full name is written in parentheses. In particular, because of the characteristics of IJMS, where Materials and Methods is arranged at the end of the paper, the original words and abbreviations are written in the order they are used from the introduction, and only when the abbreviation is used repeatedly, the abbreviation can be used until the conclusion. 

3.   Line 47: Define DMH here. As mentioned above, the abstract and the main text are separate, so only when using the abbreviation repeatedly from Introduction, you need to write the full name first, mark the abbreviation, and then use only the abbreviation until Conclusion.

4.   Line 57: Define TNFα here.

5.   Line 59: Define IL10 here.

6.   Line 64: Define TJ here.

7.   Lines 92, 108, 251, 264, more: There are still several places where IL10 is marked as IL-10, so find them all and fix them.

8.   Line 381: 0.5% and 0.1% - The space between number and unit is the problem. Determine whether or not to put a space between the unit and the number in front of the temperature unit, %, etc., and mark them all in the same pattern.

Overall, the manuscript can be considered to publication after minor revision as indicated above.

Author Response

Dear Sir/Madam,

We thank you for your insightful and helpful comments. We have addressed each of these with revisions to the text of the manuscript, accordingly

Point 1: Some additional revisions are required as pointed out below.

  1. Line 3 (Title of the Article): Two words such as 'tumor' and 'healthy' should be written as 'Tumor' and 'Healthy'.

Response 1. We have corrected it, accordingly

  1. Abbreviations: The use of abbreviations when writing a paper has many advantages besides simplicity of expression. To use an abbreviation, first write the abbreviation in parentheses after the full name, and then use the abbreviation from Introduction to the final conclusion. In this paper, in most cases, the abbreviation is written first and then the full name is written in parentheses. In particular, because of the characteristics of IJMS, where Materials and Methods is arranged at the end of the paper, the original words and abbreviations are written in the order they are used from the introduction, and only when the abbreviation is used repeatedly, the abbreviation can be used until the conclusion.

Response 2. We have taken into account this comment and  will use it as a guide for  our work in the future, thanks for the detailed explanation. 

  1. Line 47: Define DMH here. As mentioned above, the abstract and the main text are separate, so only when using the abbreviation repeatedly from Introduction, you need to write the full name first, mark the abbreviation, and then use only the abbreviation until Conclusion.
  2. Line 57: Define TNFα here.
  3. Line 59: Define IL10 here.
  4. Line 64: Define TJ here.
  5. Lines 92, 108, 251, 264, more: There are still several places where IL10 is marked as IL-10, so find them all and fix them.
  6. Line 381: 0.5% and 0.1% - The space between number and unit is the problem. Determine
    whether or not to put a space between the unit and the number in front of the temperature unit, %, etc., and mark them all in the same pattern.

Responses 3-8: We have corrected all, accordingly.

  1. The current revised manuscript has so many corrections here and there that is very difficult to read. I would like to review the revised manuscript without tracks if it is needed.

Response 9: We thank you for your attention to our work and a detailed review of the  text. We once again checked the spelling, found several more mistakes and corrected them.

Point 10: The manuscript can be considered to publication after minor revision as indicated above.

Response 10: Thank you for your helpful and positive evaluation.

Reviewer 3 Report

I accept this version 

Author Response

Dear Sir/Madam,

Thank you for your helpful and positive evaluation.

Reviewer 4 Report

Dear Authors,
your article “Selective Role of TNFα and IL10 in Regulation of Barrier Properties of 2 the Colon in DMH-induced tumor and healthy Rats” (ijms-2031339) is now a more complete work, with a comprehensive introduction,
well designed and explained experiments and a more descriptive conclusion.

Author Response

Dear Sir/Madam,

Thank you for your helpful and positive evaluation.

Point 1. English language and style are fine/minor spell check required.

Response 1. We have checked the text again and corrected a few mistakes. We thank you for your critical comments and questions, they helped us to better present our work.